# Retrospective observational study of emergency department syndromic surveillance data during air pollution episodes across London and Paris in 2014

Helen E Hughes,[1,2] Roger Morbey,[1] Anne Fouillet,[3] Céline Caserio-Schönemann,[3] Alec Dobney,[4] Thomas C Hughes,[5,6] Gillian E Smith,[1] Alex J Elliot[1]

For numbered affiliations see end of article.

**Correspondence to**
Helen E Hughes;
h.hughes7@liverpool.ac.uk

## ABSTRACT

**Introduction** Poor air quality (AQ) is a global public health issue and AQ events can span across countries. Using emergency department (ED) syndromic surveillance from England and France, we describe changes in human health indicators during periods of particularly poor AQ in London and Paris during 2014.

**Methods** Using daily AQ data for 2014, we identified three periods of poor AQ affecting both London and Paris. Anonymised near real-time ED attendance syndromic surveillance data from EDs across England and France were used to monitor the health impact of poor AQ. Using the routine English syndromic surveillance detection methods, increases in selected ED syndromic indicators (asthma, difficulty breathing and myocardial ischaemia), in total and by age, were identified and compared with periods of poor AQ in each city. Retrospective Wilcoxon-Mann-Whitney tests were used to identify significant increases in ED attendance data on days with (and up to 3 days following) poor AQ.

**Results** Almost 1.5 million ED attendances were recorded during the study period (27 February 2014 to 1 October 2014). Significant increases in ED attendances for asthma were identified around periods of poor AQ in both cities, especially in children (aged 0–14 years). Some variation was seen in Paris with a rapid increase during the first AQ period in asthma attendances among children (aged 0–14 years), whereas during the second period the increase was greater in adults.

**Discussion** This work demonstrates the public health value of syndromic surveillance during air pollution incidents. There is potential for further cross-border harmonisation to provide Europe-wide early alerting to health impacts and improve future public health messaging to healthcare services to provide warning of increases in demand.

## INTRODUCTION
### Air quality
Air pollution has negative impacts on human health. Short-term exposure to poor air quality (AQ) can affect lung function,

### Strengths and limitations of this study

► Routinely collected syndromic surveillance data from both England (London) and France (Paris) were analysed using similar health indicators.
► A single statistical method, designed specifically for daily syndromic surveillance, was applied to data from both cities.
► Air quality measurements were standardised across both cities, to overcome differences in the standard reporting from each.
► Pollutants other than particulate matter were not included, although they may be responsible for impacts on human health.
► We could not control for the potential effects of health warnings and media coverage on healthcare seeking behaviour.

including exacerbating asthma symptoms, and is associated with other acute deteriorations in respiratory and cardiovascular health.[1] Similar health effects have also been reported due to long-term exposure, with exposure to ambient air pollution associated with lung cancer and chronic respiratory and cardiovascular conditions.[1] In addition to illness within the community and increased need for healthcare, air pollution is also associated with increased mortality, with an estimated 4.7% of deaths in the England attributed to air pollution[2] and 9% of deaths in France attributed to particulate matter $(PM_{2.5})$.[3]

AQ monitoring identifies long-term trends informing policy, provides evidence of meeting (or missing) statutory target levels and quantifies the impact of preventative measures.[1,4] Daily AQ monitoring enables daily reporting of both actual and modelled AQ (predicting one or more days

in advance), for whole countries and/or individual cities, as well as on a smaller scale around individual monitoring stations.[5–7] This information is increasingly easy to access through websites and apps and is often reported through the media, especially following formal health warnings.[8]

### Syndromic surveillance

Syndromic surveillance initially focused on infectious diseases such as influenza, but is increasingly being used for other non-infectious public health events. This type of surveillance uses real-time data from patient contacts with healthcare services (eg, telephone helplines, general practice/family doctors or emergency departments (EDs)). Patient contacts/attendances are grouped by diagnoses/symptoms creating syndromic indicators such as 'respiratory' or 'gastrointestinal', providing valuable information for public health action.[9] The use of ED data lends itself particularly well to the syndromic surveillance of non-infectious public health events, with patients seeking attention for a range of acute conditions.[10–12] Previous investigation of periods of poor AQ have shown associated increases in health seeking behaviour as evidenced by syndromic surveillance, particularly for asthma and/or difficulty breathing and heart failure,[13–15] although not for myocardial infarction.[15]

### Aims

During March and early April 2014, there was a period of widespread poor AQ across Europe. In particular, the urban conurbations of London (England) and Paris (France) were affected by high temperatures, Saharan dust and industrial emissions, resulting in widespread media attention.[16–18] Here, we use routine ED syndromic surveillance data collected across London and Paris during poor AQ periods throughout 2014 to investigate the compatibility of the two countries' ED syndromic surveillance systems and describe the public health impact and associated short-term changes in healthcare seeking behaviour for selected respiratory and cardiac syndromes across different age groups.

### METHODS

### Air quality data

The area studied here has been limited to London and the whole Paris region (Île-de-France), rather than a country level. In England, the Department for Environment, Food and Rural Affairs monitors and reports on levels of air pollution using monitoring stations and provides health advice using the Daily Air Quality Index (DAQI).[8] AQ in the Paris region is monitored by Airparif and reported using the Citeair index.[19]

Both DAQI and Citeair systems monitor and report on multiple pollutants; however, each index is reported using different methodology. Therefore, the daily pollution levels across both London and Paris were standardised here, using the reported levels of $PM_{2.5}$ and $PM_{10}$. The city-wide average value for each PM on each calendar day was calculated as a mean of the maximum values reported for each monitoring station on that day, in that city.[20 21] Periods of poor AQ were then defined as those when the calculated $PM_{2.5}$ and/or calculated $PM_{10}$ average value corresponded to the DAQI index levels of 7–10, which are the PM levels classified as 'high' to 'very high' ($PM_{2.5} \geq 54\,\mu g/m^3$ and/or $PM_{10} \geq 76\,\mu g/m^3$). At these levels people, including those with no pre-existing medical conditions, are advised to consider reducing their activity levels, particularly outdoors.[7]

### Emergency department syndromic surveillance data

The Emergency Department Syndromic Surveillance System (EDSSS), is a sentinel ED system coordinated by Public Health England (PHE), collecting anonymised data from participating EDs on a daily basis (data for the previous day 00:00 to 23:59 hours are transferred to PHE the following morning).[22] Diagnosis coding in EDs in England was not standardised at the time of this investigation. Each ED had a list of diagnosis terms created locally which was available for selection in the patient attendance record. These diagnostic terms have associated codes linked to them with each ED using one of three codesets: Commissioning Data Set (CDS) Accident and Emergency Diagnosis Tables,[23] 10th revision of the International Statistical Classification of Diseases and Related Health Problems (ICD-10)[24] or Snomed CT.[25] EDs eligible for inclusion in this study were defined as those reporting using ICD-10 or Snomed CT diagnosis coding systems, which provide the level of detail required for the identification of the indicators of interest; EDs using the CDS coding system were not able to provide the coded diagnosis data in this detail. This investigation included five eligible EDSSS participating EDs in London (all located within the London PHE Centre).

The French national ED syndromic surveillance system collects daily data from the Organisation de la Surveillance COordonnée des URgences (OSCOUR) network of EDs, coordinated by Santé Publique France[26] (data for the previous day 00:00 to 23:59 hours are transferred and analysed the following morning for 85% of attendances at participating OSCOUR EDs. The OSCOUR system allows for updates and delayed reporting, the missing 15% of ED attendances from OSCOUR EDs are reported in the following 2 days[27]). All EDs reporting to OSCOUR use ICD-10 for the coding of diagnoses selected in the patient attendance record.[27] Aggregated, anonymised daily data for the Paris region (including 58 eligible EDs) were made available for this analysis.

### Epidemiological analysis

Syndromic indicators (asthma, difficulty breathing and myocardial ischaemia (MI) (table 1)) were selected from the comparable indicators already created for each system, based on clinical knowledge and experience of the potential health effects linked to air pollution and those used in previous syndromic surveillance work.

**Table 1** Syndromic surveillance indicators included in the EDSSS (London) and OSCOUR (Paris) emergency department systems and used in the study

| EDSSS (London) | OSCOUR (Paris) | Reported here as |
|---|---|---|
| Asthma | Asthme | Asthma |
| Wheeze/difficulty breathing | Dyspnée/Insuffisance respiratoire | Difficulty breathing |
| Myocardial ischaemia (MI) | Ischémie myocardique | MI |

EDSSS, Emergency Department Syndromic Surveillance System; OSCOUR, Organisation de la Surveillance COordonnée des URgences.

These syndromic surveillance indicators, which are routinely used in both EDSSS and OSCOUR, are an aggregation of relevant diagnostic codes representing similar diagnostic terms which are available in the patient record. These 'diagnoses' may not be confirmed or final and may be based on the symptoms presented, with no level of certainty indicated. The overall asthma and MI indicator groupings were very similar in each system, with the terms included all describing either asthma or MI conditions. Differences were found in non-asthma difficulty breathing-type indicators; EDSSS included symptomatic wheeze/difficulty breathing-type diagnoses, whereas OSCOUR included dyspnoea/respiratory failure diagnoses (table 2). It is worth noting that not every code listed was reported by—or even available for selection from—every ED. More relevant codes may exist for each indicator than described here; however, only codes reported to EDSSS/OSCOUR in this study are included. Although each system was found to include different codes and even numbers of codes within each indicator, they would identify most of the same patients for inclusion within the indicators used here.

For each syndromic surveillance system, attendances were aggregated by age group defined as 0–14, 15–44, 45–64 and 65 years and over.

The epidemiological analysis of ED attendance data included construction of trends in attendances for each syndromic indicator, both for all ages and for each age group, and city. The daily percentage(s) of attendances for each indicator were calculated using the number of attendances within an indicator (numerator) and the daily number of total (all-cause) attendances with a diagnosis code within each surveillance system (denominator).

### Statistical analysis

The EDSSS and OSCOUR are both live public health surveillance systems prospectively collecting data with automated contemporaneous statistical algorithms underpinning the detection of unusual activity. We applied the routine syndromic surveillance statistical detection algorithm from England: the RAMMIE method (Rising Activity, Multi-level Mixed effects Indicator Emphasis[28]). RAMMIE was applied to both English and French ED data, including to age-specific data. Using RAMMIE, two separate statistical thresholds were calculated: a '2-year' threshold (based on the previous 2 years of data) to identify significant activity compared with previous years, and a '2-week' threshold (based on the previous 2 weeks) to

identify recent, statistically significant, increases in daily activity. RAMMIE routinely allows for the prioritisation of alarms to facilitate the identification of significant activity; however, this function was not used here to ensure that all statistically significant activity was identified, and not just those signals prioritised by RAMMIE.

To ensure that sufficient data were included here to cover each of the AQ events identified, a study period of a minimum of 7 days pre the first and 7 days post the final period of poor AQ identified in London/Paris during 2014 was selected. A further period of 2 years of data prior to the first AQ event provided required baseline data for the RAMMIE method.

In addition to the RAMMIE analysis, the Wilcoxon-Mann-Whitney test was used to test for significant differences in the syndromic indicators during the 2014 study period, by age group between those days with a poor AQ and those without. To allow for the possibility of a delayed response, separate analyses were conducted incorporating lags of 1–3 days following a day of poor AQ. All analyses were undertaken using Stata V.13.1.[29]

## RESULTS
### Air quality events
During 2014, several periods of poor AQ were identified where the 'high' or 'very high' air pollution thresholds for $PM_{2.5}$ and/or $PM_{10}$ had been breached in both London and Paris (figure 1). Periods of poor AQ in Paris were generally observed to be of a longer duration and with higher DAQI levels than in London, although more individual days of poor AQ were identified in London. Two main periods of poor AQ overlapped in these cities in mid-March and early April. AQ1 was the largest event in both locations and where transboundary dust from the Sahara contributed to the makeup of the PM fraction.[13] AQ2 was apparent mainly in London (although a 1 day $PM_{10}$ spike in Paris was recorded). A third, less severe period occurred in both cities during September within a 7-day period (AQ3; table 3).

An overall study period was defined as 27 February 2014 to 1 October 2014 (216 days), to encompass each period where poor AQ occurred in both London and Paris, including 7 days before and after the first and final AQ events identified (table 3).

### ED attendances
Over the study period, 1 436 163 ED attendances were recorded across both London and Paris (table 4). Total

**Table 2** Diagnostic codes mapped to for syndromic surveillance indicators included in the EDSSS (London) and OSCOUR (Paris) emergency department systems and used in the study

| EDSSS | | | OSCOUR | |
|---|---|---|---|---|
| **Indicator** | **Code system** | **Codes** | **Indicator** | **Codes (ICD-10)** |
| Asthma | ICD-10<br>Snomed CT | J450, J459<br>30352005, 31387002, 55570000, 57546000, 161527007, 182728008, 195967001, 266364000, 281239006, 304527002, 312453004, 370204008, 370218001, 370219009, 389145006, 401135008, 409663006, 425969006, 445427006, 201031000000108, 340901000000107, 589241000000104, 653751000000109 | Asthme (Asthma) | J45, J450, J451, J458, J459, J46 |
| Difficulty breathing/ wheeze | ICD-10<br>Snomed CT | R06.0, R060, R062, R068<br>9763007, 18197001, 23141003, 24612001, 55442000, 56018004, 58596002, 60845006, 62744007, 68095009, 70407001, 161941007, 161947006, 162891007, 162894004, 230145002, 233683003, 267036007, 301703002, 301826004, 307487006, 386813002, 427354000, 427679007, 442025000, 276191000000107, 498001000000107, 498011000000109, 502631000000100, 572661000000100, 755581000000101, 755591000000104, 755611000000107, 756081000000102 | Dyspnée/Insuffisance respiratoire (Dyspnoea/ respiratory failure) | J960, J961, J961+0, J961+1, J969, R060 |
| Myocardial ischemia (MI) | ICD-10<br>Snomed CT | I200, I209, I219, I2510<br>22298006, 48447003, 53741008, 54329005, 57054005, 59021001, 67682002, 73795002, 155308009, 194828000, 233819005, 233822007, 233843008, 394659003, 398274000, 401303003, 401314000, 414545008, 414795007, 671571000000105 | Ischémie myocardique (MI) | I20, I200, I200+0, I201, I208, I209, I21, I210, I2100, I21000, I2108, I211, I2110, I21100, I2118, I2, I212, I2120, I21200, I2128, I213, I2130, I21300, I2138, I214, I2140, I21400, I2148, I219, I2190, I21900, I2198, I22, I220, I2200, I22000, I2208, I221, I2210, I22100, I2218, I228, I2280, I22800, I2288, I229, I2290, I22900, I2298, I23, I230, I231, I232, I233, I234, I235, I236, I238, I24, I240, I241, I248, I249, I25, I250, I251, I252, I253, I254, I255, I256, I258, I259 |

EDSSS, Emergency Department Syndromic Surveillance System; ICD-10, 10th revision of the International Statistical Classification of Diseases and Related Health Problems; OSCOUR, Organisation de la Surveillance COordonnée des URgences.

attendances were higher in Paris (1 163 353; from 58 EDs; >75% of all ED attendances in Paris[30]) than London (272 810; from 5 EDs, 3 using ICD-10, 2 using Snomed CT; <25% of all ED attendances in London). A comparable level of diagnosis coding was included in each city with 79% of London attendances and 72% of Paris attendances including a clinical diagnosis code.

On a weekly basis, total ED attendances in both London and Paris showed similar trends, with a peak observed on a Monday. Examination of indicator trends illustrated that there were further similarities between EDSSS and OSCOUR with highest levels of asthma attendances (as a percentage of attendances with a diagnosis code); and lowest levels of MI attendances, reported on Sundays (figure 2).

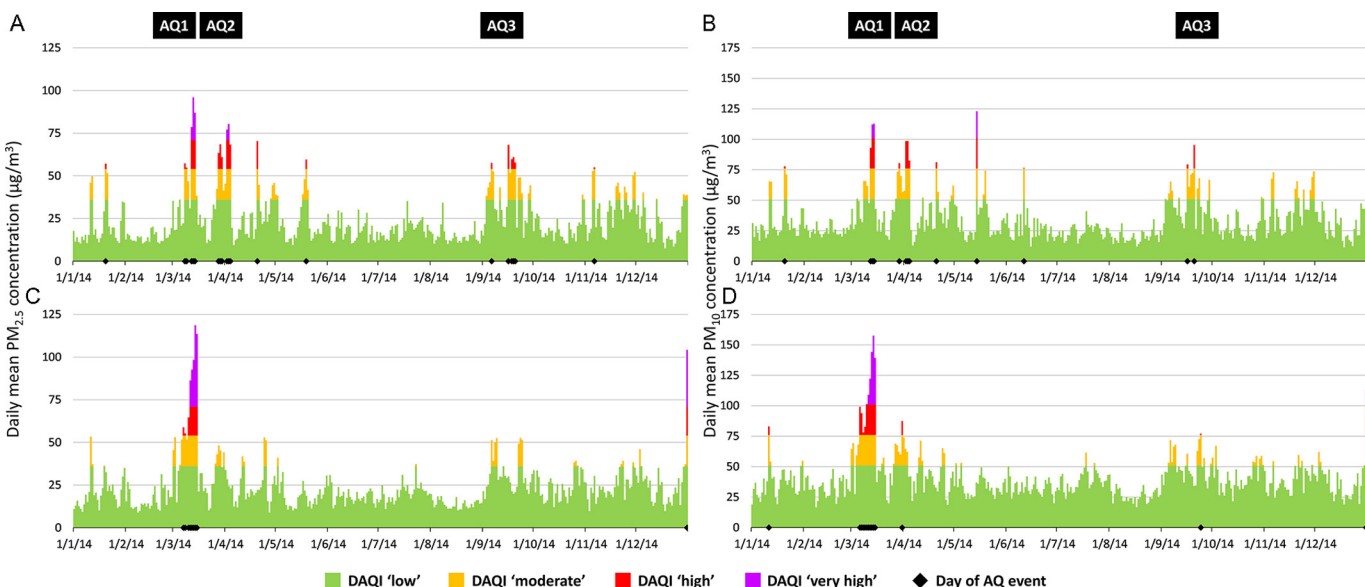

**Figure 1** Calculated mean daily particulate matter (PM) value and corresponding Daily Air Quality Index (DAQI) band, by day during 2014 in London (A) $PM_{2.5}$, (B) $PM_{10}$: Paris, (C) $PM_{2.5}$, (D) $PM_{10}$. AQ, air quality.

## ED attendances during poor air quality periods

### London ED attendances

Small increases in asthma attendances (all ages) in London EDs were observed following AQ1 (figure 3A). ED asthma attendances continued to increase during and immediately following AQ2. RAMMIE 2-week alarms were reported for the increases in asthma (all ages) immediately following AQ1 in London, indicating an attendance level higher than the previous 2 weeks. However, single 2-week alarms were not unusual in these data and were also observed during periods with no reported AQ issues. Two-year asthma alarms were not observed in the all-ages asthma attendances data during the study period.

The observed increase of asthma attendances during the AQ2 episode in London was most evident in children aged 0–14 years, and young adults (15–44 years) with each age group reaching a peak in attendances 1–2 days later (figure 3B). Asthma attendances for older adults showed no evidence of increase around periods of poor AQ (data not shown).

An additional peak in asthma (all ages) attendances was observed on 20 July 2014 (figure 3A), particularly in children (0–14 years; figure 3B), although there was no poor AQ identified at that time. During early September, increases in all-age attendances for asthma, largely driven by child attendances (0–14 years), were observed to have started prior to AQ3.

A small increase in difficulty breathing attendances (all ages) immediately following AQ2 (figure 3C) was most apparent in the older adults (aged 65 and over years; figure 3D). This single day peak was the highest level seen in this age group, around double the usual level, although not significantly higher than historical data. Other age groups were not affected.

MI attendances were less common than asthma attendances in London EDs (table 4) and affected the adult age groups almost exclusively, as would be expected. Although a peak (resulting in both 2-week and 2-year alarms) in MI attendances was observed during AQ2, particularly in those aged 65 years and over, a similar peak also occurred in late September, several days prior to the AQ3. Two-week alarms occurred quite frequently throughout the year (figure 3E,F).

### Paris ED attendances

Clear increases in ED attendances (all ages) for asthma occurred during both AQ1 and AQ2 in Paris (figure 4A) and were statistically significant (2-year and 2-week alarms). However, when broken down by age, the increase in asthma attendances in the 0–14 years age group occurred during AQ1, but not AQ2; while asthma attendances in young adults (15–44 years) were greater during AQ2 than AQ1. No statistical alarms were observed for asthma in children around AQ2, although they were present for young adults (figure 4B).

The largest peak in asthma attendances was observed on 20 July 2014, for all ages apart from 65 years and over (data not shown), matching the spike seen in London,

**Table 3** Dates of poor air quality (AQ), coinciding in London and Paris during 2014

|  | AQ1 | AQ2 | AQ3 | Total AQ days |
|---|---|---|---|---|
| London | 8 March 2014 to 14 March 2014 | 28 March 2014 to 4 April 2014 | 16 September 2014 to 20 September 2014 | 15 |
| Paris | 6 March 2014 to 15 March 2014 | 31 March 2014 | 24 September 2014 | 12 |

**Table 4** Attendances recorded in EDs, by city, over the study period (27 February 2014 to 1 October 2014)

| City | EDs | ED attendances | | | Attendances with a diagnosis | | | Indicator attendances | | |
|---|---|---|---|---|---|---|---|---|---|---|
| | | ICD-10 | Snomed CT | Total | ICD-10 | Snomed CT | Total | Asthma | Difficulty breathing | Myocardial ischaemia |
| London | 5* | 115 539 | 157 271 | 272 810 | 81 980 (71%) | 132 750 (84%) | 214 730 (79%) | 1893 (0.9%) | 812 (0.4%) | 1370 (0.6%) |
| Paris | 58 | 1 163 353 | – | 1 163 353 | 840 309 (72%) | – | 840 309 (72%) | 12 374 (1.5%) | 5433 (0.6%) | 1685 (0.2%) |

*One small ED (which used ICD-10) stopped reporting to EDSSS on 10 September 2014. All five EDs were included in descriptive and RAMMIE analysis; four EDs that reported throughout were included in Wilcoxon test and Mann-Whitney U test.
ED, emergency department; EDSSS, Emergency Department Syndromic Surveillance System; ICD-10, 10th revision of the International Statistical Classification of Diseases and Related Health Problems; RAMMIE, Rising Activity, Multi-level Mixed effects Indicator Emphasis.

despite AQ not being identified as poor on that day. One further peak in asthma attendances, apparent in all ages and individual age groups, was observed on 9–10 June 2014 (figure 4A,B). The observed peaks were not concomitant with any period of poor AQ in Paris, nor London.

Similar to London, an increase in asthma attendances was observed in Paris at the beginning of September, prior to AQ3, driven predominantly by children (0–14 years).

Difficulty breathing attendances in Paris were much lower than for asthma overall, with a single increase after AQ2 (figure 4C). Within the 15–44 years age group there was, however an increase in difficulty breathing attendances following AQ1 (figure 4D).

Attendances for MI in Paris showed no evidence of increase during/following days of poor AQ (figure 4E,F), although some statistical alarms were observed throughout the year, particularly a series of three 2-year alarms during late August and September in those aged 65 years and over.

### Retrospective statistical analysis

Wilcoxon-Mann-Whitney test results provide further evidence, alongside the descriptive epidemiology and RAMMIE results, that there is a strong association between days of poor AQ and asthma attendances for all ages and particularly in children aged 0–14 years (table 5). Furthermore, the statistical significances of the associations between asthma attendances and poor AQ were highest when modelled with a lag between the day of poor AQ and attendances; 2 days for London and 3 days for Paris. Although there was some evidence of increased attendances for difficulty breathing and MI in some age groups in London 1 day after poor AQ, these alarms were single significant values (rather than the grouping of significant asthma results by age group). These increased MI and difficulty breathing attendances in the day following poor AQ were not seen in the Paris data.

### DISCUSSION
### Main findings

We used two national ED syndromic surveillance systems to describe and compare the short-term changes in ED indicators during periods of poor AQ in two European capital cities. The AQ events reported here in Paris and London were related to the same pollutants ($PM_{2.5}/PM_{10}$), and were very similar in terms of the dates and duration, and changes in public health outcomes in terms of ED attendances.

The most sensitive ED indicator during periods of poor AQ was asthma, with the impact most apparent up to 3 days after a day of poor AQ. The breakdown of attendances by age group revealed some differences, with the strongest associations overall seen between poor AQ and asthma attendances in children. This finding was consistent with previous studies, which have shown children to

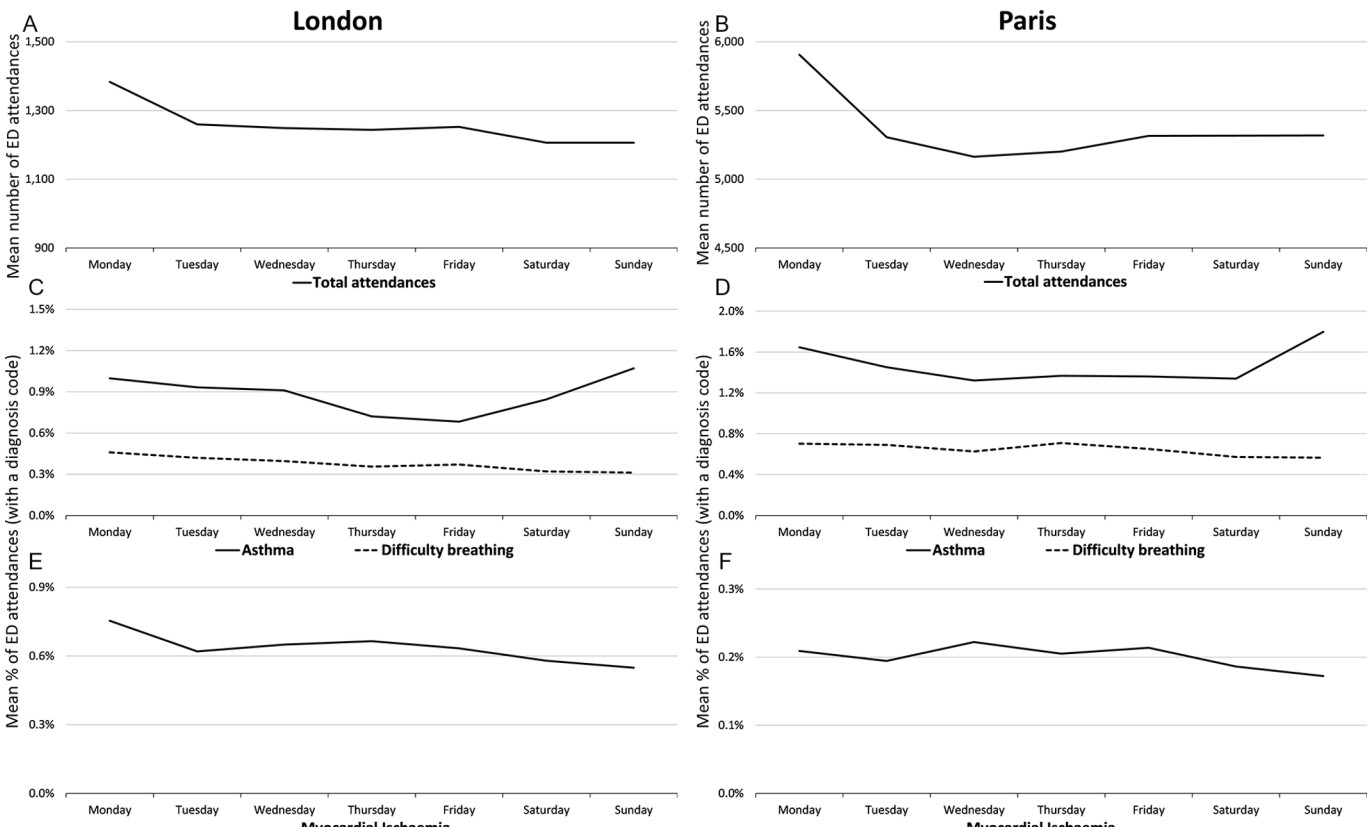

**Figure 2** Mean emergency department (ED) attendances by day of week, 27 February 2014 to 1 October 2014, by syndromic indicators, London reported to Emergency Department Syndromic Surveillance System ((A) total attendances, (C) asthma and difficulty breathing, (E) myocardial ischaemia) and Paris reported to Organisation de la Surveillance COordonnée des URgences ((B) total attendances, (D) asthma and difficulty breathing, (F) myocardial ischaemia).

be more susceptible to exacerbation of asthma symptoms requiring healthcare in association with air pollution.[31]

The investigation of individual AQ incidents demonstrated the potential for differing levels of impact on different age groups at different times. Although generally children were most affected by AQ, a large increase in adult asthma attendances was observed during and immediately following AQ2 in both London and Paris. Within England, this increase in attendances around AQ2 has previously been described.[32] As the second period of poor AQ to occur in a short period of time, media coverage and the associated communication of health warning information and interventions put in place during AQ2 may have resulted in changes in behaviour which affected the levels of exposure of different age groups.

In addition to the increases observed during AQ periods, a sharp increase in asthma attendances (all ages) was observed in Paris on 9–10 June 2014, and in both London and Paris on 20 July 2014. These peaks did not coincide with any AQ event identified here; however, additional meteorological data (not presented) revealed periods of major thunderstorm activity within each city at the time.[33–35] These findings match those previously reported, including from the EDSSS, describing the health effects of 'thunderstorm asthma', where sudden exacerbation of asthma symptoms results in increased healthcare seeking behaviour over a short time period,[12 36–39] possibly due to

increased levels of pollen and fungal spores, although the mechanism has not yet been confirmed.[36]

We also observed further increases in asthma attendances in both Paris and London towards the start of September. This increase was particularly evident in children and is likely linked to an annual 'back to school' increase in asthma-type attendances in EDs during September.[40–42]

Other syndromic indicators investigated showed little (difficulty breathing) to no (MI) association with the AQ incidents identified here.

### Strengths and limitations

The OSCOUR system includes greater representative coverage nationally, with more EDs participating than the sentinel EDSSS system (540 EDs across France were reporting to OSCOUR).[30] While 34 EDs across England and Northern Ireland were reporting to EDSSS at 20 March 2014, the five reported here were located in London making the EDSSS more representative in London than at the national level.[43] The large number of OSCOUR EDs reported here resulted in much more stable data from Paris, reducing background noise and allowing clearer differentiation of spikes/increases in attendances. The smaller number of attendances within the EDSSS data made identifying spikes 'harder'; however, the use of RAMMIE enables significant increases

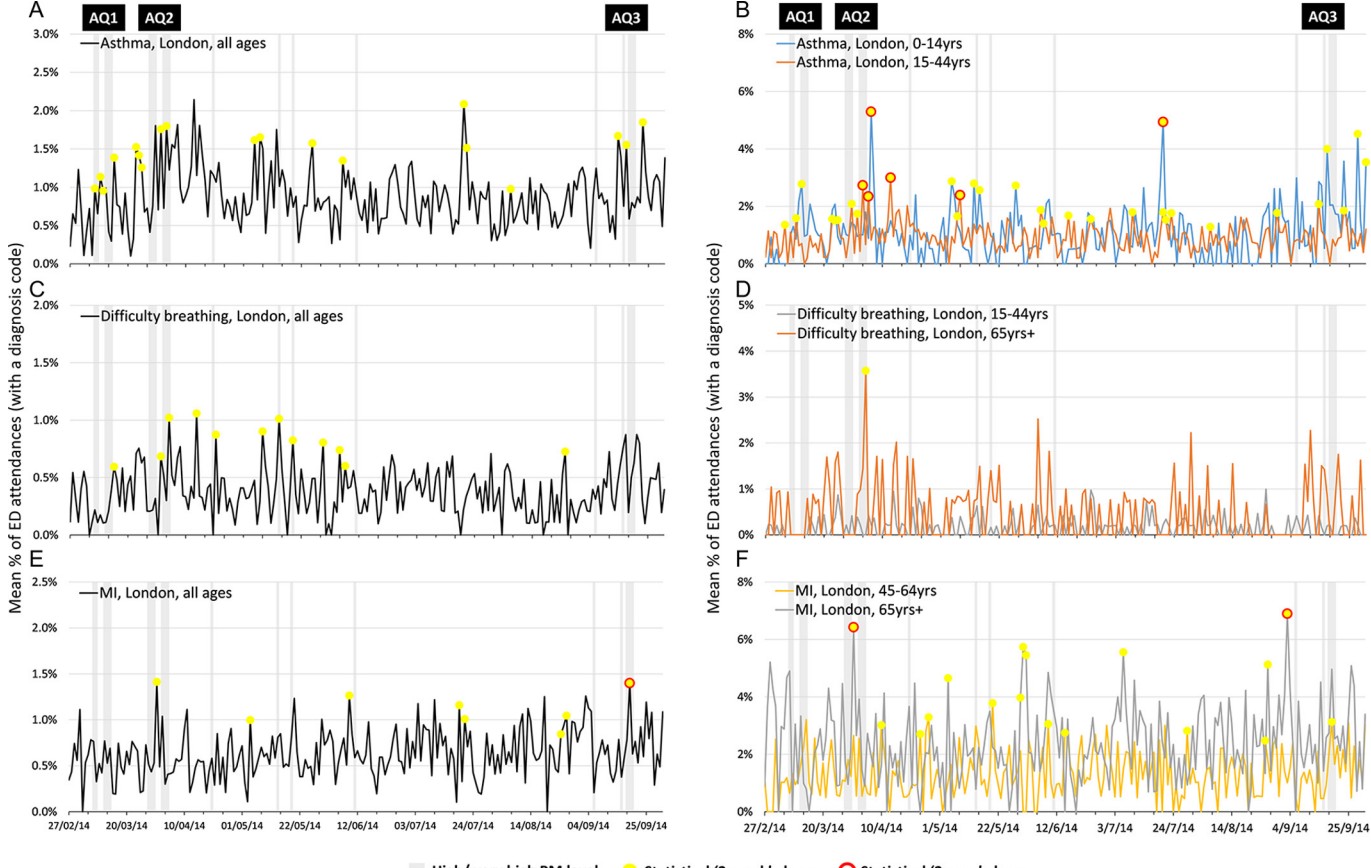

**Figure 3** Daily percentages of London emergency department (ED) attendances for syndromic surveillance indicators of (A) asthma all ages, (B) asthma 0–14 and 15–44 years, (C) difficulty breathing all ages, (D) difficulty breathing 15–44 and 65+ years, (E) myocardial ischaemia (MI) all ages and (F) MI 45–64 and 65+ years, with statistical alarms, reported to Emergency Department Syndromic Surveillance System. AQ, air quality; PM, particulate matter.

in attendances to be identified, even when not initially obvious.[28]

Despite underlying differences in the method of data collection, with EDSSS taking a single snapshot of daily attendances and OSCOUR allowing the initial snapshot data to be updated retrospectively, both systems reported over 70% completion of the clinical diagnosis field making diagnostic data comparable. Furthermore, although these systems were developed individually, it was found that the syndromic indicators used within each system were similar, making comparisons of health impact possible. However, the EDSSS used a wheeze/difficulty breathing indicator, whereas OSCOUR used a difficulty breathing/respiratory failure indicator. This difference is, in part, likely to be related to the use of different clinical coding systems, with the identification of symptoms (eg, wheeze) more difficult using ICD-10 (as used in France) than Snomed CT (used by some EDs in England).

The percentage of ED visits (with a diagnosis code), as an indication of ED attendances, as reported here (rather than actual numbers) may be impacted by the overall levels of ED attendances (and levels of diagnostic coding) on any 1 day. Although travel and outdoor activities are discouraged during AQ events, there are other factors which have a much greater impact on

ED attendances (such as national and school holiday periods). The patterns and total numbers of attendances during 2014, including AQ periods, were not different from those seen in other years. The normal levels of overall ED attendances observed during periods of poor AQ, although travel was discouraged, contrasts with the reduced overall ED attendances in the English EDSSS seen during extreme cold weather when transportation is not physically possible for most people.[10] By using percentage of attendances, the impact of events, such as periods of poor AQ, can be clearly seen in terms of changes in ED workload, such as changes in case mix and/or age groups attending.

The levels of attendances for each indicator were different between cities, with respiratory indicators higher in Paris (asthma 1.5%, difficulty breathing 0.7%), than London (asthma 0.9%, difficulty breathing 0.4%) and MI attendances higher in London (0.6%) than in Paris (0.2%). This disparity in attendance levels between countries may be due to differences in diagnosis coding practices, clinical procedures used for treating patients (eg, immediate transfer to cardiac care rather than ED for patients with MI) or even areas of specialty for each ED (eg, some London EDs are part of specialist heart care hospitals so may see more patients with MI). However, the

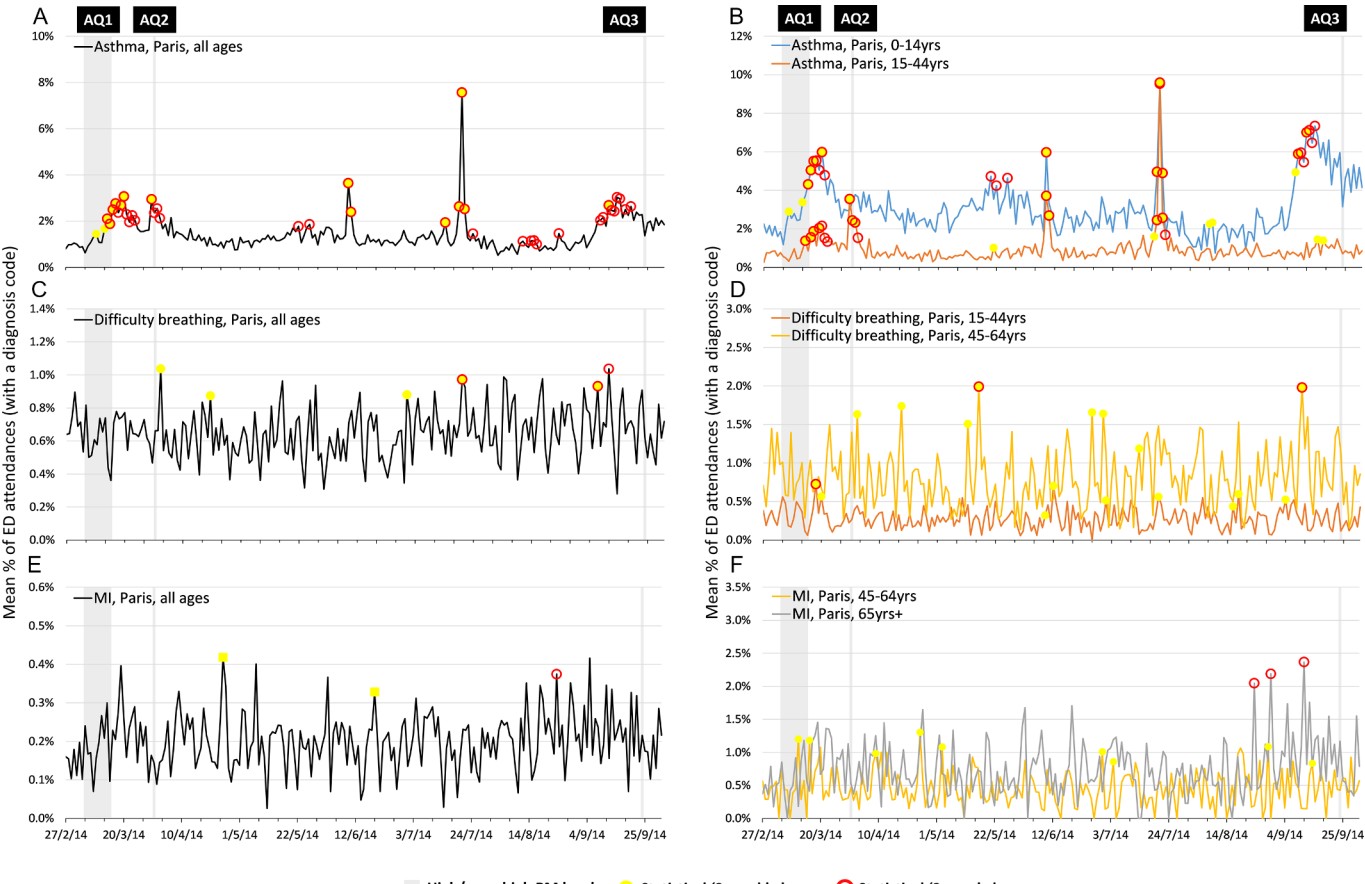

**Figure 4** Daily percentages of Paris emergency department (ED) attendances for syndromic surveillance indicators of (A) asthma all ages, (B) asthma 0–14 and 15–44 years, (C) difficulty breathing all ages, (D) difficulty breathing 15–44 and 45–64 years, (E) myocardial ischaemia all ages and (F) myocardial ischaemia 45–64 and 65+ years, with statistical alarms, reported to OSCOUR. AQ, air quality; PM, particulate matter.

trends observed within weeks were very similar in both systems, implying they are broadly comparable (figure 2).

A limitation of the statistical methods used here is that the occurrence of previous events (eg, poor AQ or weather systems) influencing the indicators were not identified or removed from the 2 years of historical data used as RAMMIE training data. The potential inclusion of unrecognised events may impact on the RAMMIE model thresholds, although 2 years is considered sufficient for meaningful results (R. Morbey, personal communication).

This study focused solely on PM, although other pollutants impact on human health. The application of the DAQI levels to both London and Paris mean daily data allowed for an international comparison, based on days with higher than usual $PM_{2.5}$ and/or $PM_{10}$ specific to each city. The use of the highest daily $PM_{2.5}/PM_{10}$ values was considered, but these values were found to be high/very high on the DAQI scale on the majority of days of 2014.

The impact of health warnings and media reporting associated with actual and predicted periods of poor AQ could not be controlled for here. The intention of health warnings, which are reported in the media, is to reduce the impact on human health, encouraging the public to reduce exposure as recommended.[8 44] There were increases in asthma attendances in children during and following AQ1 in Paris in particular, although these younger age groups appeared unaffected during later events, whereas young adults were more greatly affected by AQ2. These differences of impact by age group in AQ2 may have been due to changes in behaviour of younger age groups so soon after AQ1 and subsequent reduced exposure to poor AQ, rather than a biological response observed in adults only. In addition to the impact of media reporting, France has introduced several other measures when AQ limit values are exceeded in major cities; speed limits, alternate driving days (to limit the number of cars on the road) and free public transportation. The implementation of these measures could have had an impact on the results presented here.

It is important here to underline that variations of near real-time indicators are not easy to attribute directly to poor AQ. An absence of short term variation (eg, MI in this study) cannot not be interpreted as a total lack of any long-term impact. Similarly, the identification of a significant increase in syndromic indicators reported here (eg, asthma) has not formally accounted for other associated factors such as climatic conditions (eg, weather and allergens) or viral circulation. Further time series analysis should be completed to control for potential confounding factors.

**Table 5** Results of the Wilcoxon test and Mann-Whitney U test illustrating the standardised value (z value) and significance (P value) of syndromic indicators to days of poor air quality (including 1–3 day lag).

| Indicator | City | Lag (days) | All ages z value | All ages P value | 0–14 years z value | 0–14 years P value | 15–44 years z value | 15–44 years P value | 46–64 years z value | 46–64 years P value | 65+ years z value | 65+ years P value |
|---|---|---|---|---|---|---|---|---|---|---|---|---|
| Asthma | London | 0 | −0.227 | 0.8204 | −2.857 | **0.0043** | 1.287 | 0.1982 | 1.077 | 0.2813 | −1.009 | 0.3128 |
| | | 1 | −1.443 | 0.1490 | −3.213 | **0.0013** | 0.556 | 0.5784 | −0.791 | 0.4291 | −1.026 | 0.3048 |
| | | 2 | −1.713 | **0.0867** | −3.838 | **0.0001** | 0.787 | 0.4310 | −0.558 | 0.5768 | −1.438 | 0.1503 |
| | | 3 | −1.627 | 0.1038 | −2.574 | **0.0100** | −0.141 | 0.8876 | −0.442 | 0.6586 | 0.816 | 0.4145 |
| | Paris | 0 | −0.963 | 0.3356 | −1.566 | 0.1173 | 0.529 | 0.5971 | −0.624 | 0.5326 | 0.000 | 1.0000 |
| | | 1 | −2.035 | **0.0419** | −2.576 | **0.0100** | −0.330 | 0.7418 | −1.582 | 0.1137 | −0.354 | 0.7237 |
| | | 2 | −2.706 | **0.0068** | −3.090 | **0.0020** | −0.943 | 0.3454 | −2.558 | **0.0105** | −0.194 | 0.8464 |
| | | 3 | −3.049 | **0.0023** | −3.201 | **0.0014** | −1.797 | **0.0724** | −2.77 | **0.0056** | −0.756 | 0.4499 |
| Difficulty breathing | London | 0 | −0.055 | 0.9563 | −0.963 | 0.3357 | 1.311 | 0.1898 | −0.361 | 0.7181 | −0.140 | 0.8889 |
| | | 1 | −1.261 | 0.2073 | −2.975 | **0.0029** | 1.797 | **0.0723** | 0.445 | 0.6564 | −0.728 | 0.4666 |
| | | 2 | −0.444 | 0.6573 | −1.385 | 0.1659 | 0.223 | 0.8236 | 1.452 | 0.1464 | −0.580 | 0.5620 |
| | | 3 | −1.552 | 0.1207 | −1.236 | 0.2166 | −0.695 | 0.4872 | −0.01 | 0.9916 | −0.296 | 0.7670 |
| | Paris | 0 | −0.604 | 0.5459 | 0.031 | 0.9749 | −0.585 | 0.5582 | −0.736 | 0.4615 | −0.147 | 0.8830 |
| | | 1 | −0.057 | 0.9547 | −1.032 | 0.3021 | −0.490 | 0.6242 | 0.603 | 0.5466 | −0.078 | 0.9376 |
| | | 2 | −1.364 | 0.1725 | −1.095 | 0.2735 | −0.674 | 0.5004 | −0.565 | 0.5722 | −1.521 | 0.1283 |
| | | 3 | −1.144 | 0.2526 | −0.528 | 0.5974 | −0.942 | 0.3464 | 0.427 | 0.6697 | −1.222 | 0.2217 |
| Myocardial infarction | London | 0 | −0.605 | 0.5452 | – | – | −0.084 | 0.9327 | −1.275 | 0.2022 | 0.027 | 0.9787 |
| | | 1 | −0.588 | 0.5565 | – | – | 0.329 | 0.7421 | −1.994 | **0.0461** | 0.374 | 0.7084 |
| | | 2 | −0.081 | 0.9354 | – | – | −0.084 | 0.9327 | −0.61 | 0.5419 | 0.053 | 0.9574 |
| | | 3 | −0.571 | 0.5680 | – | – | 0.544 | 0.5862 | −1.415 | 0.1571 | −0.695 | 0.4873 |
| | Paris | 0 | −0.364 | 0.7158 | 0.546 | 0.5850 | −1.257 | 0.2089 | −0.089 | 0.9293 | 0.367 | 0.7138 |
| | | 1 | 0.243 | 0.8082 | 0.546 | 0.5850 | −1.257 | 0.2089 | −0.022 | 0.9828 | 1.594 | 0.1110 |
| | | 2 | −0.331 | 0.7408 | 0.546 | 0.5850 | −0.522 | 0.6016 | −0.235 | 0.8141 | 0.635 | 0.5253 |
| | | 3 | −0.676 | 0.4992 | 0.546 | 0.5850 | −0.578 | 0.5630 | 0.384 | 0.7011 | −0.403 | 0.6872 |

Figures in bold are significant to the 90% significance level; those bold and underlined to the 95% significance level.

## FUTURE WORK

This work has prompted the systematic investigation of asthma attendances by age group around AQ events by Public Health England, using the EDSSS. In France (following the March 2014 periods of poor AQ reported here), the health authorities requested and are now provided with, systematic surveillance of OSCOUR ED attendances for asthma by age group during poor AQ events. This work shows the potential of real-time syndromic surveillance to enhance the public health response to air pollution incidents, even if real-time changes observed through syndromic surveillance data cannot be absolutely related to air pollution. As the evidence base for the utility of syndromic surveillance during air pollution events increases, it is hoped that it will, in combination with environmental data, be used by authorities to provide public health messaging during future events: messages to the public to advise about risks and preventative measures, and to EDs and other health service providers about increases in patient numbers and changes to the case mix of patients attending.

The increases in attendance levels for specified indicators, particularly asthma in children, provides an insight into the age groups affected, and how the workload and case mix within EDs can rapidly change. Contemporaneous feedback may be given on the utility of health warnings issued which may aid in the targeting of advice to particular age groups and also the preparations made in EDs in terms of staffing and materials required.

Where increased ED attendances were observed during periods of no known changes in AQ, there is potential for further investigation of the potential causes. The identification of periods of thunderstorm activity on the days of the highest asthma attendances reported here should be investigated further.

This study is the first example of the RAMMIE method being applied to a syndromic surveillance system outside the UK, identifying and highlighting increases in ED attendances during periods of known poor AQ. This work has illustrated the potential for RAMMIE to be applied to countries developing new syndromic surveillance systems, or without the infrastructure to support bespoke statistical developments. However, the limitations of this method must always be considered, where increased levels resulting in statistical alarms (either 2 weeks or 2 years) must be viewed alongside local intelligence and

knowledge, not every alarm will be due to poor AQ, but the indicators can be used for monitoring the impact of AQ events on public health.

This work also promotes further collaboration between different countries to explore methods to harmonise syndromic surveillance systems. Other public health surveillance initiatives have been adopted across Europe to provide a means of reporting singularly comparable variables and statistics across several countries, including: the European monitoring of excess mortality for public health action[45]; the European Influenza Surveillance Scheme[46]; establishment of epidemic thresholds for influenza surveillance[47]; the European Antimicrobial Resistance Surveillance Network[48]; harmonised norovirus surveillance systems also exist.[49 50] Within this study, although ED indicators were not entirely harmonised, they had been developed to be the most appropriate for each system and country. This work has also stimulated opportunities to explore other areas of public health that could be enhanced using a multinational syndromic surveillance system in particular those due to non-infectious causes such as injury surveillance and these will be addressed in future work.

The apparent difference in the noise-to-signal ratio between OSCOUR and EDSSS, that is, background variation was likely due to the size of each respective network. Peaks of abnormal activity were easier to identify in OSCOUR and therefore future work within PHE is currently focusing on expanding the EDSSS to improve its geographical representativeness and increase the attendance numbers, thereby reducing the noise-to-signal ratio.

The potential for the harmonisation of syndromic surveillance across national borders is also clear, with opportunities to build on local experience to bring international public health benefits.

### Author affiliations

[1]Real-time Syndromic Surveillance Team, National Infection Service, Public Health England, Birmingham, UK
[2]The Farr Institute, The Health eResearch Centre, University of Liverpool, Liverpool, UK
[3]Syndromic Surveillance Unit, Santé Publique France, The National Public Health Agency, Paris, France
[4]Environmental Hazards and Emergencies Department, Centre for Radiation, Chemical and Environmental Hazards, Public Health England, Birmingham, UK
[5]Emergency Department, John Radcliffe Hospital, Oxford, UK
[6]The Royal College of Emergency Medicine, London, UK

**Acknowledgements** The authors acknowledge the contribution and support from the ED clinicians and Trust staff in the EDSSS; the ongoing support of the Royal College of Emergency Medicine; the technical support provided by EMIS Health and L2S2 Ltd in developing the EDSSS. The authors also acknowledge the contribution from the ED structures and clinicians involved in the OSCOUR network in France; the ongoing support of the Federation of the Regional Observatories of Emergencies; the Scientific Society of Emergency Medicine; the air and climate team of Santé publique France and the Regional Unit of Santé publique France in Paris-Ile-de-France region. The authors acknowledge and thank Sylvia Medina (Santé Publique France) for professional input and critical revision of this manuscript. HEH receives support from the National Institute for Health Research Health Protection Research Unit (NIHR HPRU) in Gastrointestinal Infections. AJE and GES receive support from the National Institute for Health Research Health Protection Research Unit (NIHR HPRU) in Emergency Preparedness and Response.

The views expressed are those of the author(s) and not necessarily those of the NHS, the NIHR, the Department of Health or Public Health England.

**Contributors** HEH contributed to the study design, prepared the ED data for England, completed the statistical analyses, drafted the manuscript and provided critical revision and final approval of the manuscript. RM contributed to the study design, completed the statistical analyses, drafted the manuscript and provided critical revision and final approval of the manuscript. AF contributed to the study design, prepared the ED data for France and provided critical revision and final approval of the manuscript. AD contributed to the study design, prepared the air quality data and provided critical revision and final approval of the manuscript. CC-S, TCH, GES, AJE contributed to the study design, critical revision and final approval of the manuscript.

**Funding** This research received no specific grant from any funding agency, commercial or not-for-profit sectors. This surveillance is undertaken as part of the national surveillance functions of Public Health England and Santé Publique France.

**Competing interests** None declared.

**Patient consent** Not required.

**Ethics approval** Ethical approval for this work was not required. The anonymised EDSSS health data used in this study were routinely collected at part of the public health function of PHE. The collection and analysis of data provided by the OSCOUR network in the frame of public health surveillance and epidemiological studies has been authorised by the French National Commission for Data protection and Liberties (CNIL).

**Provenance and peer review** Not commissioned; externally peer reviewed.

**Data sharing statement** Additional data are not available for sharing.

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
