## [Reviewer comments · BMJ Open]

ARTICLE DETAILS

TITLE (PROVISIONAL)	'A tale of two cities': a comparison of emergency department syndromic surveillance data during air pollution episodes across London and Paris in 2014
AUTHORS	Hughes, Helen Morbey, Roger Fouillet, Anne Caserio-Schönemann, Céline Dobney, Alec Hughes, Thomas Smith, Gillian Elliot, Alex

VERSION 1 - REVIEW

REVIEWER	howard burkom
REVIEW RETURNED	07-Sep-2017

GENERAL COMMENTS	Thank you for the opportunity to review this manuscript. At a high level, the authors achieve their goals of illustrating two neighboring surveillance operations. Illustration of similarities and differences are interesting, and study limitations are described well. A good follow-up study would be to examine the operations to quantify transmission between countries, even without data from neighboring localities, because there is likely a high volume of daily travel between Paris and London. I request that the authors make modifications to address the points below. 1. On p. 7, line 24, the authors say that "The syndromic surveillance indicators available were similar in both EDSSS and OSCOUR®, with minor differences only in nonasthma difficulty breathing type conditions." I do not get this impression from Supplementary Table 1. If I read the table correctly, the EDSSS system uses fewer ICD-10 codes but supplements them with SNOMED codes. Please discuss the differences in more detail.2. Most readers will not keep clear the difference between spike alarms and historical alarms, and the term "spike" has other typical usages (e.g. "matching the spike seen in London" on p. 12). I would prefer "2-week alarms" and "2-year alarms". Also, for the 2-year alarms, please clarify whether percentages for the entire 2 years are used to calculate the baseline.3. On p. 6, line 34, does "(index levels of 7-10: PM2.5 \geq54 or PM10 \geq76)" correspond only to "very high" or to both "high" and "very high"?4. From the referenced RAMMIE article, please make brief reference to the prioritization rules and indicate whether they were applied.5. The discussion of algorithm results is somewhat loose and needs to be more precise. On p. 13, line 36, it should not be claimed that
--

“Wilcoxon-Mann-Whitney testing confirmed the descriptive epidemiology and RAMMIE results”. I agree that strong associations were found between days of poor AQ and elevated RAMMIE statistics as a result of this testing (depending on the next question). Similarly, I think that the statement “Historical asthma alarms are less frequent and were not observed in these data during the study period” refers only to all-age data. Please review the algorithm results analysis.

6. In the application of the Wilcoxon-Mann-Whitney test, were the z-scores (and separately p-values) ranked and then the statistic calculated according to whether days were classified as poor AQ? Please state the number of poor AQ days and total days in each city.

7. Please replace the unclear statement on p. 10, line 20:

“Overlaying these alarms on the daily percentage of attendances for each syndromic indicator and the periods of poor AQ showed where significantly higher than expected levels of ED attendances were observed.”

Here is one suggestion: “Figures 1 and 2 show plots of daily attendance percentages along with (vertically shaded) periods of poor AQ, and the marking of RAMMIE alarms on these plots shows the degree of correspondence of poor AQ with statistically anomalous high ED attendance.”

8. On p. 7, line 54, please reword the statement “... analysis of ED attendance data included construction of trends using the daily number of total (all cause) attendances with a diagnosis code within each surveillance system (denominator) and the number of attendances within an indicator (numerator).” One possibility is “... analysis of ED attendance data included monitoring the daily ratio of the number of attendances with an indicator (numerator) to the number of all cause attendances with any diagnosis code (denominator), within each surveillance system”. (or please correct this interpretation if it is wrong)

Editorial issues:

1. Plain language is preferable, but for the sake of the reader unfamiliar with the subject, please complete demonstrative pronoun subjects: p.13, line 34, “Though there was some evidence of increased attendances for difficulty breathing and MI selected age groups in London [one] day after poor AQ, these (alarms?) were single significant values; p. 13, line 57: “These (isolated alarms?) were also not reflected in the”; p. 14, line 41 “Within England this (?) has previously been described”; p. 16, line 12 “This (?) is, in part, likely to be related”; p. 16, line 26: “This (?) may be due to differences in diagnosis coding”; p. 16, line 42 “This (?) may impact on the RAMMIE model thresholds”.

2. Please review instances of the word “data” and be consistent about treating it as plural.

3. Please correct the spelling of Wilcoxon-Mann-Whitney on p.2, line 35; p.8, line 41; and p. 31, line 3 (caption)

4. Figure captions should be located with the corresponding figures and shouldn't be presented a second time alone in the text. (This problem may pertain to the journal staff rather than the authors.)

5. The text font size of all of the figures needs to be larger and in darker font. Figures with larger font sizes should still fit on the page.

6. Even with larger text, the labeling of components of Figures 1 and 2 needs to be clearer. It appears that Figures A, C, and E give all-age percentages, while B, D, and F give percentages for selected age group pairs, and different age group pairs are selected for each indicator. In the caption, please specify the indicators and the age groups for the respective letters.

7. The small circles representing the spike statistical alarms are

	difficult to distinguish on the manuscript pdf page. Please use a different symbol that will also make clear when the spike alarms co-occur with historical alarms. 8. If the figures and supplementary figures are all in the body of the text, why not simply number them as figures 1-4?
--	---

REVIEWER	Richard S. Hopkins, MD, MSPH
REVIEW RETURNED	25-Oct-2017

GENERAL COMMENTS	i am glad to have had a chance to review this interesting paper. Transnational comparisons are important and have not been common to date, and it is encouraging to see syndromic surveillance methods applied to an environmental health problem. The stated aims have two parts: to investigate the compatibility of the two cities' surveillance systems; and to estimate public health impact of AQ events for selected syndromes, overall and by age group. The former seems to have been carried out more completely than the latter. There is no actual measurement of health impact, other than observations of statistically significant changes in ED visits for various syndromes between AQ days and other days. i was looking for something like an estimate of the proportion of ED visits by syndrome that could be attributed to AQ events, either for the whole study period or for the duration of each AQ event's impact (allowing for latent periods of a few days). One well-known weakness of displaying syndrome data as 'percent of daily visits' is that it can go up when visits for other causes go down. This should be addressed as a potential weakness. For example, during AQ events, if people are being encouraged to stay home, not drive their cars, etc, then visits to EDs for reasons other than respiratory symptoms might actually go down, and the percentage with respiratory diagnoses might go up further. Did total ED visits change during AQ event periods? While this project uses data collected using a syndromic surveillance mechanism, with daily submission of information about all ED visits, it is really diagnostic data, relying heavily on ICD-10-coded data, rather than syndromic data and should be identified as such. I can't tell how much of the English data is based on combinations of SNOMED codes, which are not diagnostic in the same sense. On a related matter: how soon after each day's visit are the ICD-10-coded data about those visits available for analysis? Comments in the paper suggest that for the data from Paris, there might be a delay (how many days?) while additional data are added to records, while the English data are a snapshot of each day's data. For greatest public health utility, the data should be available in as close to real time as possible. If one used the French data the day after the ED visits whose data are sought, how much of what this paper relies on would be missing? In supplemental Table 1, the lists ICD-10 codes used by the two countries' systems to assign visits to syndrome categories are compared. Based on the text, I was expecting the codes for asthma and for MI to be more similar between countries than they are. Are the differences only in codes that occur rarely in the respective systems, thus bolstering the idea that the lists are really similar? I
--

	have no way to tell. For example, for asthma, the English system has two codes while the French system has six, including the two used in England. How many records are added because of the presence of one of the other four codes? For MI, the French list is much longer than the English. Also, is there reason to think that French and English ED staff use the ICD10 codes similarly for similar illnesses? What is known about the comparability of the almost 30% of records without a diagnostic code to those with such codes? For example, were there spikes in 'no-diagnosis' visits on the same days as spikes in asthma? If so, this would suggest that a fair number of true asthma visits may be in the no-diagnosis records. Spikes in the number of no-diagnosis records on other days would suggest that in fact the target conditions are not predominant among those records. This goes to inferences from the data: one wants to say that a spike in ED visits represents a spike in illness in the population, with the ED visits captured in the syndromic surveillance systems representing a subset of all the illnesses that have occurred. If one knew the actual diagnoses or syndromes of the no-diagnosis visits, would it change one's assessment of the overall picture? I would like to see a tally of alert signals for asthma in relation to AQ spikes. There is discussion of such signals on days that turned out to be thunderstormy but not AQ event days. If one relied on these data alone to identify AQ events (which of course in the real world is not necessary -- you only have to look out the window to get an idea), how often would one be wrong with respect to PM 1.0 or 2.5? Of course if the spikes in asthma visits are being driven by thunderstorms, that is a 'real' cause as well. That suggests another type of analysis of the data -- how often are thunderstorm days followed by spikes in asthma visits to EDs? -- outside the scope of this paper, I realize. There was a recent spectacular asthma spike, with associated deaths, in Melbourne, Australia, related to strong thunderstorms. I have not seen reports of what that looks like in data from Melbourne's syndromic surveillance network. Finally, how do French and English public health authorities plan to use these syndromic surveillance system data to assist with planning for and response to future adverse air quality events, now that they understand the properties of the system?
--	--

REVIEWER	Samuel L. Groseclose
REVIEW RETURNED	29-Oct-2017

GENERAL COMMENTS	Note: This review represents the opinions of the reviewer and does not represent the official position of the Centers for Disease Control and Prevention. Very well-written manuscript clearly states the research objectives, the methods used to harmonize air quality measures, and the syndromic surveillance systems and data analyzed. Very comprehensive discussion of Strengths and Limitations of analysis. Page 7, lines 24-31: Description of case definitions could be clarified a bit. I understand that EDSSS uses both SNOMED and ICD-10 codes and OSCOUR uses ICD-10 only. But curious to know why the same ICD-10 syndrome code were not used where they could be
---

	used by the two systems? A brief description of how ICD-10 and SNOMED codes are used in EDSSS would be helpful. Per your description, it seems that some EDs use ICD-10 and others us only SNOMED? Has there been a comparison of how asthma, for example, is represented as a health outcome in ICD vs SNOMED? Authors do mention the impact of coding systems in limitations section (p.16). Page 12, lines 8-10 and 24-29 – Description of increases in asthma preceding AQ3 and MI peak preceding AQ3 are not mentioned in the Results section. Authors should discuss these results in addition to those that aligned with poor AQ periods. If this system were to be implemented routinely and data used to inform risk communication, these findings should be investigated further (e.g., mentioned in Future Work section).
--	--

VERSION 1 – AUTHOR RESPONSE

Reviewer 1

Thank you for the opportunity to review this manuscript. At a high level, the authors achieve their goals of illustrating two neighboring surveillance operations. Illustration of similarities and differences are interesting, and study limitations are described well. A good follow-up study would be to examine the operations to quantify transmission between countries, even without data from neighboring localities, because there is likely a high volume of daily travel between Paris and London. I request that the authors make modifications to address the points below.

Response: we thank the reviewer for their positive comments and we will certainly consider a follow-up study as suggested.

1. On p. 7, line 24, the authors say that “The syndromic surveillance indicators available were similar in both EDSSS and OSCOUR®, with minor differences only in nonasthma difficulty breathing type conditions.” I do not get this impression from Supplementary Table 1. If I read the table correctly, the EDSSS system uses fewer ICD-10 codes but supplements them with SNOMED codes. Please discuss the differences in more detail.

Response: The text in the ‘methods: epidemiological analysis’ section has been made clearer with regards to the codes used for analysis: ‘The overall asthma and MI indicator groupings were very similar in each system, with the terms included all describing either asthma or myocardial ischaemic conditions. Differences were found in non-asthma difficulty breathing type indicators, where EDSSS included symptomatic wheeze/ difficulty breathing type diagnoses and OSCOUR® included dyspnoea/ respiratory failure diagnoses (supplementary table 1). Please note: not every code listed was reported by – or even available for selection – from every ED. More relevant codes may exist for each indicator than described here, however only codes reported to EDSSS/ OSCOUR® in this study are included’.

2. Most readers will not keep clear the difference between spike alarms and historical alarms, and the term “spike” has other typical usages (e.g. “matching the spike seen in London” on p. 12). I would prefer “2-week alarms” and “2-year alarms”. Also, for the 2-year alarms, please clarify whether percentages for the entire 2 years are used to calculate the baseline.

Response: thank you, this has been changed.

3. On p. 6, line 34, does “(index levels of 7-10: PM2.5 \geq 54 or PM10 \geq 76)” correspond only to “very high” or to both “high” and “very high”?

Response: this has been made clearer in the text. ‘Periods of poor AQ were then defined as those when either the calculated PM2.5 or PM10 average value corresponded to the DAQI index levels of 7-10, which are classified as ‘high’, to ‘very high’ (PM2.5 \geq 54 $\mu\text{g}/\text{m}^3$ or PM10 \geq 76 $\mu\text{g}/\text{m}^3$).’

4. From the referenced RAMMIE article, please make brief reference to the prioritization rules and indicate whether they were applied.

Response: This information has been included in the methods. ‘RAMMIE routinely allows for the prioritisation of alarms to facilitate the identification of significant activity, however, this function was

not used here to ensure that all statistically significant activity was identified, and not just those signals prioritised by RAMMIE'

5. The discussion of algorithm results is somewhat loose and needs to be more precise. On p. 13, line 36, it should not be claimed that "Wilcoxon-Mann-Whitney testing confirmed the descriptive epidemiology and RAMMIE results". I agree that strong associations were found between days of poor AQ and elevated RAMMIE statistics as a result of this testing (depending on the next question). Similarly, I think that the statement "Historical asthma alarms are less frequent and were not observed in these data during the study period" refers only to all-age data. Please review the algorithm results analysis.

Response: We have clarified this paragraph to make it more precise, as requested. 'Wilcoxon-Mann-Whitney test results provide further evidence, alongside the descriptive epidemiology and RAMMIE results, that there is a strong association between days of poor AQ and asthma attendances'

6. In the application of the Wilcoxon-Mann-Whitney test, were the z-scores (and separately p-values) ranked and then the statistic calculated according to whether days were classified as poor AQ? Please state the number of poor AQ days and total days in each city.

Response: We applied the Wilcoxon-Mann-Whitney test to identify whether when ED attendances for e.g. asthma, were ranked these ranks were significantly associated with whether or not the days included poor AQ. To be clear, we did not use the Wilcoxon Mann Whitney tests to rank the number of RAMMIE alarms each day or to rank any RAMMIE generated daily statistics of significance. The numbers of days (in total and AQ days) have been included in table 2

7. Please replace the unclear statement on p. 10, line 20: "Overlaying these alarms on the daily percentage of attendances for each syndromic indicator and the periods of poor AQ showed where significantly higher than expected levels of ED attendances were observed."

Here is one suggestion: "Figures 1 and 2 show plots of daily attendance percentages along with (vertically shaded) periods of poor AQ, and the marking of RAMMIE alarms on these plots shows the degree of correspondence of poor AQ with statistically anomalous high ED attendance."

Response: this has been reworded, thank you. 'RAMMIE alarms, where significantly higher than expected levels of ED attendances were observed, showed a degree of correspondence with the dates of poor AQ (figures 3 & 4)'

8. On p. 7, line 54, please reword the statement "... analysis of ED attendance data included construction of trends using the daily number of total (all cause) attendances with a diagnosis code within each surveillance system (denominator) and the number of attendances within an indicator (numerator)." One possibility is "... analysis of ED attendance data included monitoring the daily ratio of the number of attendances with an indicator (numerator) to the number of all cause attendances with any diagnosis code (denominator), within each surveillance system". (or please correct this interpretation if it is wrong)

Response: this has been reworded, thank you. 'The epidemiological analysis of ED attendance data included construction of trends in attendances for each syndromic indicator, both for all ages and for each age group, and city. The daily percentage of attendances for each indicator were calculated using the number of ²attendances within an indicator (numerator) and the daily number of total (all cause) attendances with a diagnosis code within each surveillance system (denominator)'

Editorial issues:

1. Plain language is preferable, but for the sake of the reader unfamiliar with the subject, please complete demonstrative pronoun subjects: p.13, line 34," Though there was some evidence of increased attendances for difficulty breathing and MI selected age groups in London [one] day after poor AQ, these (alarms?) were single significant values; p. 13, line 57: "These (isolated alarms?) were also not reflected in the"; p. 14, line 41 "Within England this (?) has previously been described"; p. 16, line 12 "This (?) is, in part, likely to be related"; p. 16, line 26: "This (?) may be due to differences in diagnosis coding"; p. 16, line 42 "This (?) may impact on the RAMMIE model thresholds".

Response: these have been reworded and other examples also corrected

2. Please review instances of the word "data" and be consistent about treating it as plural.

Response: this has been corrected

3. Please correct the spelling of Wilcoxon-Mann-Whitney on p.2, line 35; p.8, line 41; and p. 31, line 3 (caption)

Response: thank you, this has been corrected.

4. Figure captions should be located with the corresponding figures and shouldn't be presented a second time alone in the text. (This problem may pertain to the journal staff rather than the authors.)

Response: Figure captions have been moved from within the text, to after the references

5. The text font size of all of the figures needs to be larger and in darker font. Figures with larger font sizes should still fit on the page.

Response: charts have been reformatted

6. Even with larger text, the labeling of components of Figures 1 and 2 needs to be clearer. It appears that Figures A, C, and E give all-age percentages, while B, D, and F give percentages for selected age group pairs, and different age group pairs are selected for each indicator. In the caption, please specify the indicators and the age groups for the respective letters.

Response: Charts have been reformatted to make differences clearer

7. The small circles representing the spike statistical alarms are difficult to distinguish on the manuscript pdf page. Please use a different symbol that will also make clear when the spike alarms co-occur with historical alarms.

Response: charts have been reformatted to make this clearer

8. If the figures and supplementary figures are all in the body of the text, why not simply number them as figures 1-4?

Response: this has been corrected

Reviewer: 2

I am glad to have had a chance to review this interesting paper. Transnational comparisons are important and have not been common to date, and it is encouraging to see syndromic surveillance methods applied to an environmental health problem.

Response: we thank the reviewer for their positive comments.

The stated aims have two parts: to investigate the compatibility of the two cities' surveillance systems; and to estimate public health impact of AQ events for selected syndromes, overall and by age group. The former seems to have been carried out more completely than the latter. There is no actual measurement of health impact, other than observations of statistically significant changes in ED visits for various syndromes between AQ days and other days. I was looking for something like an estimate of the proportion of ED visits by syndrome that could be attributed to AQ events, either for the whole study period or for the duration of each AQ event's impact (allowing for latent periods of a few days).

Response: Although we have demonstrated a statistically significant increase in ED attendances across both cities, we are not able, from this piece of work to provide an estimate of how many ED visits this may relate to for any one AQ event. This may be possible in future work. We have replaced the word 'estimate' with 'describe' in the aims section

One well-known weakness of displaying syndrome data as 'percent of daily visits' is that it can go up when visits for other causes go down. This should be addressed as a potential weakness. For example, during AQ events, if people are being encouraged to stay home, not drive their cars, etc, then visits to EDs for reasons other than respiratory symptoms might actually go down, and the percentage with respiratory diagnoses might go up further. Did total ED visits change during AQ event periods?

Response: There is always a difficult decision required as to how to present syndromic ED data, in the absence of complete data from a national system, we have chosen to present percentages. This has been added and addressed as a potential weakness. 'The use of percentage or ED visits (with a diagnosis code), as an indication of ED attendances (rather than actual numbers), as reported here may be impacted by the overall levels of ED attendances (and levels of diagnostic coding) on any one day. Though travel and outdoor activities are discouraged during AQ events, there are other factors which have a much greater impact on ED attendances (such as national and school holiday periods). The patterns and total numbers of attendances during 2014, including AQ periods, were not different to those seen in other years. This is different to the changes seen during extreme cold weather when

total attendances has been seen in to be reduced in the English EDSSS, as transportation is not physically possible for most people [11]. By using percentage of attendances the impact of events, such as periods of poor AQ, can be clearly seen in terms of changes in ED workload, such as changes in case mix and/ or age groups attending.'

While this project uses data collected using a syndromic surveillance mechanism, with daily submission of information about all ED visits, it is really diagnostic data, relying heavily on ICD-10-coded data, rather than syndromic data and should be identified as such. I can't tell how much of the English data is based on combinations of SNOMED codes, which are not diagnostic in the same sense.

Response: This has been addressed in the epidemiological analysis section of the Methods: 'Though 'diagnostic' information these diagnoses have potentially been made before any final confirmation and may be based on the symptoms presented, with no level of certainty indicated.'

Table 3 has been extended (provided separately from the main document) to provide a breakdown of total attendances and the number of attendances with a diagnosis code included by the coding system (ICD-10/SnomedCT). A breakdown by indicator has not been provided as this is likely to be more greatly influenced by areas of specialty of the ED, rather than the coding system in use. The description of indicators also now makes it clear that MI and asthma indicators in all cases include named diagnoses, whereas the difficulty breathing indicator in EDSSS allows for symptomatic codes: 'The overall asthma and MI indicator groupings were very similar in each system, with the terms included all describing either asthma or myocardial ischaemic conditions. Differences were found in non-asthma difficulty breathing type indicators, where EDSSS included symptomatic wheeze/ difficulty breathing type diagnoses and OSCOUR® included dyspnoea/ respiratory failure diagnoses'.

On a related matter: how soon after each day's visit are the ICD-10-coded data about those visits available for analysis? Comments in the paper suggest that for the data from Paris, there might be a delay (how many days?) while additional data are added to records, while the English data are a snapshot of each day's data. For greatest public health utility, the data should be available in as close to real time as possible. If one used the French data the day after the ED visits whose data are sought, how much of what this paper relies on would be missing?

Response: we agree with the reviewers comment there is a potential delay in the OSCOUR system - for both systems data are available for analysis the day following each ED visit. The OSCOUR® system does allow for updates on this, though 85% of records are analysed on the day following the ED visit. 'data for the previous day 00:00 to 23:59 are transferred and analysed the following morning for 85% of ED attendances. OSCOUR® allowing for updates and delayed reporting, the missing 15% of ED attendances are reported in the following 2 days'

In supplemental Table 1, the lists ICD-10 codes used by the two countries' systems to assign visits to syndrome categories are compared. Based on the text, I was expecting the codes for asthma and for MI to be more similar between countries than they are. Are the differences only in codes that occur rarely in the respective systems, thus bolstering the idea that the lists are really similar? I have no way to tell. For example, for asthma, the English system has two codes while the French system has six, including the two used in England. How many records are added because of the presence of one of the other four codes? For MI, the French list is much longer than the English. Also, is there reason to think that French and English ED staff use the ICD10 codes similarly for similar illnesses?

Response: this comment is similar to reviewer 1, which we have addressed above.

What is known about the comparability of the almost 30% of records without a diagnostic code to those with such codes? For example, were there spikes in 'no-diagnosis' visits on the same days as spikes in asthma? If so, this would suggest that a fair number of true asthma visits may be in the no-diagnosis records. Spikes in the number of no-diagnosis records on other days would suggest that in fact the target conditions are not predominant among those records. This goes to inferences from the data: one wants to say that a spike in ED visits represents a spike in illness in the population, with the ED visits captured in the syndromic surveillance systems representing a subset of all the illnesses that have occurred. If one knew the actual diagnoses or syndromes of the no-diagnosis visits, would it change one's assessment of the overall picture?

Response: Unfortunately we hold no information on the attendances which do not have an assigned diagnosis code. Some may have left the ED before investigation/ treatment, some may still be awaiting diagnosis at the time the data was extracted. It is assumed that the practices within EDs are consistent on any given day and that there is no bias in the inclusion of a diagnosis in the data

extract. In this way trends over time may be estimated, rather than a definitive calculation of the numbers of people affected.

I would like to see a tally of alert signals for asthma in relation to AQ spikes. There is discussion of such signals on days that turned out to be thunderstormy but not AQ event days. If one relied on these data alone to identify AQ events (which of course in the real world is not necessary -- you only have to look out the window to get an idea), how often would one be wrong with respect to PM 1.0 or 2.5? Of course if the spikes in asthma visits are being driven by thunderstorms, that is a 'real' cause as well. That suggests another type of analysis of the data -- how often are thunderstorm days followed by spikes in asthma visits to EDs? -- outside the scope of this paper, I realize. There was a recent spectacular asthma spike, with associated deaths, in Melbourne, Australia, related to strong thunderstorms. I have not seen reports of what that looks like in data from Melbourne's syndromic surveillance network.

Response: It is a limitation of this study that a single potential cause (high particulate matter levels) was included in the analysis. Other air quality indicators may also have an impact detectable in the EDs, though the longer term processes involved were thought to make this unlikely. Thunderstorms are of particular interest and were only identified following the completion of the analysis, similarly the increase in asthma attendances following the school summer holidays had a very clear impact in both countries. Syndromic surveillance is regularly used to identify events (when of course, there may be false 'alarms' due to chance, unrecognised agents/ events), but holds great value in the monitoring of impact levels and where possible reassurance that no impact has been observed. The ability for RAMMIE to monitor and reassure has been demonstrated here. 'However, the limitations of this method must always be considered, where increased levels resulting in statistical alarms (either 2-week or 2-year) must be viewed alongside local intelligence and knowledge, not every alarms will be due to poor AQ, but the indicators can be used for monitoring the impact of AQ events on public health.'

Finally, how do French and English public health authorities plan to use these syndromic surveillance system data to assist with planning for and response to future adverse air quality events, now that they understand the properties of the system?

Response: This work has prompted the systematic investigation of asthma attendances by age group around AQ events in England and Northern Ireland, using the EDSSS and in France using OSCOUR®. This detail has now been given in the 'future work' section. 'This work has prompted the systematic investigation of asthma attendances by age group around AQ events in England and Northern Ireland, using the EDSSS. In France, since the March 2014 periods of major AQ problems reported here, the health authorities requested and are now provided with, systematic surveillance of OSCOUR® ED attendances for asthma by age group during poor quality events.'

Reviewer: 3

Very well-written manuscript clearly states the research objectives, the methods used to harmonize air quality measures, and the syndromic surveillance systems and data analyzed. Very comprehensive discussion of Strengths and Limitations of analysis.

Response: we thank the reviewer for their positive comments.

Page 7, lines 24-31: Description of case definitions could be clarified a bit. I understand that EDSSS uses both SNOMED and ICD-10 codes and OSCOUR uses ICD-10 only. But curious to know why the same ICD-10 syndrome code were not used where they could be used by the two systems? A brief description of how ICD-10 and SNOMED codes are used in EDSSS would be helpful. Per your description, it seems that some EDs use ICD-10 and others us only SNOMED? Has there been a comparison of how asthma, for example, is represented as a health outcome in ICD vs SNOMED? Authors do mention the impact of coding systems in limitations section (p.16).

Response: this comment is similar to reviewer 1 & 2, which we have addressed above.

Page 12, lines 8-10 and 24-29 – Description of increases in asthma preceding AQ3 and MI peak preceding AQ3 are not mentioned in the Results section. Authors should discuss these results in addition to those that aligned with poor AQ periods. If this system were to be implemented routinely and data used to inform risk communication, these findings should be investigated further (e.g., mentioned in Future Work section).

Response: increased asthma attendances prior to AQ3 in both London and Paris are included in the results, though has been made more explicit in the Paris results, MI alarms have now also been mentioned in more detail. The future work section has been re-ordered and expanded.

VERSION 2 – REVIEW

REVIEWER	Howard Burkom
REVIEW RETURNED	24-Dec-2017

GENERAL COMMENTS	The authors have carefully addressed the criticisms from my previous review. The residual points below are minor. If the editors still require another review, the changes should be easy, and I will try to respond promptly to a revision. 1. I previously questioned the interpretation of Supplementary Table 1 and said “If I read the table correctly, the EDSSS system uses fewer ICD-10 codes but supplements them with SNOMED codes. Please discuss the differences in more detail.” I don’t think that the revision directly addresses this comment. If the authors mean to say that the two systems capture most of the same cases despite the fact that EDSSS uses fewer ICD codes, then I agree and request only that they make the text explicit on this point along with the rest of the explanation. 2. I previously questioned the statement “Historical asthma alarms are less frequent and were not observed in these data during the study period.” Based on the new figures 3 and 4, please either say “were not observed in the all-age data during the study period”, or else modify the statement. 3. I asked for the number of poor AQ days and total days in each city. The authors responded “The numbers of days (in total and AQ days) have been included in table 2”. In the version that I received, they did supply this information, but not in Table 2. The number of AQ days has been appended to Table 3, and the total number of days is in the text above that table. If the authors are satisfied, so am I. Editorial issues: 1. I requested antecedents for the demonstrative pronouns and appreciate the authors’ efforts to clarify the multiple instances. Two remain and should be easy to remedy: On p. 16, line 49 is the text “The largest peak in asthma attendances was observed on 20/07/14, ..., matching the spike seen in London, despite this not being a poor AQ period”. Please change the final clause to “despite the fact that the air quality on this date was not poor”. On p. 21, line 27 is the text “The patterns and total numbers of attendances during 2014, including AQ periods, were not different [from] those seen in other years.” (Please make the bracketed change.) “This[] is different to the changes seen during extreme cold weather when total attendances has been seen in to be reduced in the English EDSSS,”. (Please also remove extra words and/or correct the grammar—I do not understand this new clause.) 2. Please add the bracketed “s” in the revised statement The daily percentage[s] of attendances for each indicator were calculated using the number of attendances within an indicator (numerator) and the daily number of total (all cause) attendances with a diagnosis code within each surveillance system (denominator) 3. The spelling of Wilcoxon-Mann-Whitney still needs correction in the heading of Table 5; other instances are now correct.
--

	Regarding the figures: 4. I requested that the text font size of all of the figures needs to be larger and in darker font. What I see in the revision is the helpful addition of colour, but at 100% magnification, the text is still difficult to read. I needed 200% magnification to be able to read the axis labels and captions. The problem may be for the journal rather than the authors. 5. The figures on the left are for all ages, and those on the right are for pairs of age groups that the authors chose depending on the monitored condition. I ask again that the authors make these points clear, including specification of the chosen age groups, in the figure captions, especially with the figure text so difficult to read. 6. I appreciate that the authors have modified the alarm symbols to allow discernible co-occurrence of the two types of alarms. However, the symbol for the 2-week alarms is a yellow unbordered circle that may not be visible on the printed page and definitely won't be visible on some screen projections. It is a question for authors and editors whether a clearer symbol should be substituted.
--	---

REVIEWER	Richard S. Hopkins, MD, MSPH
REVIEW RETURNED	20-Dec-2017

GENERAL COMMENTS	The authors have been very diligent and complete about addressing all the reviewers' comments. Thank you. Most of my further comments below are of the copy-editing variety, not on the substance of the paper, which is in fine shape. I was, however, somewhat disappointed in their response to the suggestion that the authors elaborate on the present and future public health value of the syndromic surveillance data they used for this project. Mostly they describe further research efforts to follow up on certain observations made here. That research will doubtless be valuable, but I was hoping for some discussion of how real-time SS data on respiratory illness can be helpful to public health authorities. Would there ever be a time when public advisories would be based on SyS data plus environmental monitoring data, not just the latter? Would SyS be used to intensify or moderate such advisories? Would SyS data be used to help make staffing decisions about EDs or hospital inpatient units? On line 3 of page 3, the abstract says that this work 'demonstrates the public health value of real-time syndromic surveillance.....'. Could you add a sentence here, as well as in the discussion, that describes the nature of that value? Page 1, line 3. I miss the "Tale of Two Cities" language in the title. Page 6, line 35. I don't understand what is 'high' and what is 'very high', since you reference two different measurements (PM 2.5 and PM 10). Is it high if either or these measurements is exceeded? if so, what is 'very high'? Page 6 line 58 and page 7 line 3. This is the only mention of Commissioning Data Set Accident and Emergency Diagnosis Table. I suggest omitting it. Page 7 line 11. Those of us from the other side of the Atlantic may
--

	not know precisely what 'central London' means. Probably not the City, and probably not the whole conurbation? Page 7, line 21. Check punctuation. Page 8, lines 7-10. Suggested rewording: "...and available in the patient record. These 'diagnoses' may not be final and be based on the symptoms...." Page 8, lines 20-22. Do you mean that EDSSS included symptomatic wheeze etc and OSCOUR did not, and that OSCOUR included dyspnea and EDSSS did not? This could be clearer. Page 8, line 27. Unpaired parenthesis. page 12, lines 19-26. There is a long sentence here that should be broken into 3 or 4 shorter sentences. For example, one sentence per AQ event. Page 13, first paragraph. What % of estimated total ED attendances in each of your two regions are captured in those regions' SyS systems? page 13, lines 25-29. check punctuation page 15 lines 5 to 12 (halfway). This text repeats Methods description. page 16, line 21. New sentence should start after AQ3. Page 16, lines 31-35. Much of this material is redundant to Methods. Page 20, lines 14-16. Reference to England and France as a whole is unnecessary and distracting. Page 20, line 38. This is the first place we learn that the London data are based on FIVE EDs. This should be brought out much earlier in the paper. Page 21, line 19. Delete "use of". It is the percentages that are impacted by levels of ED attendance, not the use of the percentages.
--	--

REVIEWER	Samuel L. Groseclose
REVIEW RETURNED	18-Dec-2017

GENERAL COMMENTS	In the revised manuscript, the authors have satisfactorily addressed my initial comments on the manuscript and those of the other reviewers. I have no additional substantive comments on the manuscript as revised and recommend acceptance for publication. Nice collaborative analysis of retrospective syndromic surveillance data. Well-done description of the surveillance findings, study limitations, and recommendations for further work.
--

VERSION 2 – AUTHOR RESPONSE

Reviewer: 3

Reviewer Name: Samuel L. Groseclose

the authors have satisfactorily addressed my initial comments on the manuscript and those of the other reviewers. I have no additional substantive comments on the manuscript as revised and recommend acceptance for publication.

Nice collaborative analysis of retrospective syndromic surveillance data. Well-done description of the surveillance findings, study limitations, and recommendations for further work.

-We thank the reviewer for their positive comments.

Reviewer: 2

Reviewer Name: Richard S. Hopkins, MD, MSPH

Most of my further comments below are of the copy-editing variety, not on the substance of the paper, which is in fine shape.

- Thank you

I was, however, somewhat disappointed in their response to the suggestion that the authors elaborate on the present and future public health value of the syndromic surveillance data they used for this project. Mostly they describe further research efforts to follow up on certain observations made here. That research will doubtless be valuable, but I was hoping for some discussion of how real-time SS data on respiratory illness can be helpful to public health authorities. Would there ever be a time when public advisories would be based on SyS data plus environmental monitoring data, not just the latter? Would SyS be used to intensify or moderate such advisories? Would SyS data be used to help make staffing decisions about EDs or hospital inpatient units?

- We agree with the reviewer that ultimately it is the aim that a combination of SyS and environmental data would be used by public health authorities to a) provide health advice to the public and b) inform EDs about future pressures/changes in case mixes. However, this is not the current situation as the evidence base for the utility of SyS in air pollution incidents is small and though added to by this study needs further work. While we do not want to make 'false promises' in this paper about what SyS can deliver, we have added some further explanation of some of these future aims. We hope that this now meets the reviewers comment.

On line 3 of page 3, the abstract says that this work 'demonstrates the public health value of real-time syndromic surveillance.....'. Could you add a sentence here, as well as in the discussion, that describes the nature of that value?

- We have added the additional text to the discussion as described above. We have also amended the abstract to also reflect this.

Page 1, line 3. I miss the "Tale of Two Cities" language in the title.

- Unfortunately this had to be removed to meet the requirements of BMJ Open style guide

Page 6, line 35. I don't understand what is 'high' and what is 'very high', since you reference two different measurements (PM 2.5 and PM 10). Is it high if either or these measurements is exceeded? if so, what is 'very high'?

- For this work an AQ event was defined as when PM2.5 and/or PM10 breached the 'high' value or even higher than that, in the 'very high' range. The DAQI index uses both high and very high for measuring and reporting, we used both these levels to indicate a period of poor air quality. We hope that the changes to the text make this more clear.

Page 6 line 58 and page 7 line 3. This is the only mention of Commissioning Data Set Accident and Emergency Diagnosis Table. I suggest omitting it.

- We think that it is important to include the mention of the Commissioning Dataset for completeness – we have added a further sentence to explain why it was not used further in the analysis

Page 7 line 11. Those of us from the other side of the Atlantic may not know precisely what 'central London' means. Probably not the City, and probably not the whole conurbation?

- Unfortunately it is actually very difficult to define 'Central London', particularly as we are not able to identify the individual EDs, so we have amended this to read 'London' which we hope will be easier for non-UK-based readers to understand.

Page 7, line 21. Check punctuation.

- Corrected, thank you

Page 8, lines 7-10. Suggested rewording: "...and available in the patient record. These 'diagnoses' may not be final and be based on the symptoms...."

- Reworded and simplified, thank you

Page 8, lines 20-22. Do you mean that EDSSS included symptomatic wheeze etc and OSCOUR did not, and that OSCOUR included dyspnea and EDSSS did not? This could be clearer.

- Yes, we do mean that EDSSS includes symptomatic codes which were not included in OSCOUR. We have reworded this section to make this clearer.

Page 8, line 27. Unpaired parenthesis.

- Corrected, thank you

page 12, lines 19-26. There is a long sentence here that should be broken into 3 or 4 shorter sentences. For example, one sentence per AQ event. Done rejected deletion of the word whereas as the sentence didn't read properly. Also adjusted the new shorted sentences a little to make them read better

- We have separated these sentences, as suggested

Page 13, first paragraph. What % of estimated total ED attendances in each of your two regions are captured in those regions' SyS systems?

- We have included the coverage value for Paris (>75%) as published in the reference provided. Unfortunately this is a surprisingly difficult estimate to calculate for the London data due to ED organisation and levels of care provided. In order to provide a comparable estimate for London we have estimated the London coverage (<25%, to match the banding used in the Paris document) as the 5 EDs in London were from 4 Trusts of the 21 reported to have provided type 1 (Major) EDs during quarter 1 2014.

page 13, lines 25-29. check punctuation

- Corrected, thank you

page 15 lines 5 to 12 (halfway). This text repeats Methods description. Removed

- We thank the reviewer for this point and we have removed this unnecessary section text

page 16, line 21. New sentence should start after AQ3.

- Corrected, thank you

Page 16, lines 31-35. Much of this material is redundant to Methods.

- As above we have removed the text where repeating the methods and simplified the paragraph.

Page 20, lines 14-16. Reference to England and France as a whole is unnecessary and distracting.

- This has been removed

Page 20, line 38. This is the first place we learn that the London data are based on FIVE EDs. This should be brought out much earlier in the paper.

- Whilst this information was included in the paper (page 13 and Table 4) we have included reference to the numbers of participating EDs in the methods so that it is clearer to the reader earlier in the manuscript

Page 21, line 19. Delete "use of". It is the percentages that are impacted by levels of ED attendance, not the use of the percentages.

- Corrected, thank you

Reviewer: 1

Reviewer Name: Howard Burkom

The authors have carefully addressed the criticisms from my previous review. The residual points below are minor. If the editors still require another review, the changes should be easy, and I will try to respond promptly to a revision.

-We thank the reviewer for their time and consideration when reviewing this manuscript

1. I previously questioned the interpretation of Supplementary Table 1 and said “If I read the table correctly, the EDSSS system uses fewer ICD-10 codes but supplements them with SNOMED codes. Please discuss the differences in more detail.” I don’t think that the revision directly addresses this comment. If the authors mean to say that the two systems capture most of the same cases despite the fact that EDSSS uses fewer ICD codes, then I agree and request only that they make the text explicit on this point along with the rest of the explanation.

- Though each system includes a range of different codes (which may or may not all be reported by all EDs in that system), each does capture most of the same cases. We believe we have made this more clear in the text.

2. I previously questioned the statement “Historical asthma alarms are less frequent and were not observed in these data during the study period.” Based on the new figures 3 and 4, please either say “were not observed in the all-age data during the study period”, or else modify the statement.

- Corrected, thank you

3. I asked for the number of poor AQ days and total days in each city. The authors responded “The numbers of days (in total and AQ days) have been included in table 2”. In the version that I received, they did supply this information, but not in Table 2. The number of AQ days has been appended to Table 3, and the total number of days is in the text above that table. If the authors are satisfied, so am I

- Apologies, the correction had been made, but the wrong table number used in the response to the reviewer

Editorial issues:

1. I requested antecedents for the demonstrative pronouns and appreciate the authors’ efforts to clarify the multiple instances. Two remain and should be easy to remedy:

On p. 16, line 49 is the text “The largest peak in asthma attendances was observed on 20/07/14, ..., matching the spike seen in London, despite this not being a poor AQ period”. Please change the final clause to “despite the fact that the air quality on this date was not poor”.

- Corrected, thank you

On p. 21, line 27 is the text “The patterns and total numbers of attendances during 2014, including AQ periods, were not different [from] those seen in other years.” (Please make the bracketed change.)

- Corrected, thank you

“This[] is different to the changes seen during extreme cold weather when total attendances has been seen in to be reduced in the English EDSSS.”. (Please also remove extra words and/or correct the grammar—I do not understand this new clause.)

- Corrected, thank you

2. Please add the bracketed “s” in the revised statement The daily percentage[s] of attendances for each indicator were calculated using the number of attendances within an indicator (numerator) and the daily number of total (all cause) attendances with a diagnosis code within each surveillance system (denominator)

- Corrected, thank you

3. The spelling of Wilcoxon-Mann-Whitney still needs correction in the heading of Table 5; other instances are now correct.

- Corrected, thank you

Regarding the figures:

4. I requested that the text font size of all of the figures needs to be larger and in darker font. What I see in the revision is the helpful addition of colour, but at 100% magnification, the text is still difficult to read. I needed 200% magnification to be able to read the axis labels and captions. The problem may be for the journal rather than the authors.

- We thank the reviewer for these helpful comments – we have enlarged the in figure legends for figures 3 and 4, and will work with the journal on making sure that the figures meet the journal requirements for production

5. The figures on the left are for all ages, and those on the right are for pairs of age groups that the authors chose depending on the monitored condition. I ask again that the authors make these points clear, including specification of the chosen age groups, in the figure captions, especially with the figure text so difficult to read.

- We have added the age groups to the figure captions to clarify these points

6. I appreciate that the authors have modified the alarm symbols to allow discernible co-occurrence of the two types of alarms. However, the symbol for the 2-week alarms is a yellow unbordered circle that may not be visible on the printed page and definitely won't be visible on some screen projections. It is a question for authors and editors whether a clearer symbol should be substituted.

- We thank the reviewer for these helpful comments. We will work with the journal on making sure that the figures meet the journal requirements for production.